# Development of an Optimized Clearing Protocol to Examine Adipocyte Subpopulations in White Adipose Tissue

**DOI:** 10.3390/mps4020039

**Published:** 2021-06-02

**Authors:** Quyen Vi Luong, Andreas Israel, Rita Sharma, Siegfried Ussar, Kevin Y. Lee

**Affiliations:** 1The Diabetes Institute, Ohio University, Athens, OH 45701, USA; ql255013@ohio.edu (Q.V.L.); sharmar1@ohio.edu (R.S.); 2Department of Biomedical Sciences, Heritage College of Osteopathic Medicine, Ohio University, Athens, OH 45701, USA; 3German Center for Diabetes Research (DZD), 85764 Neuherberg, Germany; andreas.israel@helmholtz-muenchen.de (A.I.); siegfried.ussar@helmholtz-muenchen.de (S.U.); 4Department of Medicine, Technical University of Munich, 80333 Munich, Germany; 5RG Adipocytes & Metabolism, Institute for Diabetes & Obesity, Helmholtz Diabetes Center, Helmholtz Center Munich, 85764 Neuherberg, Germany

**Keywords:** adipocyte subpopulation, solvent-based clearing, confocal microscopy

## Abstract

Organic solvent dibenzyl ether (DBE)-based protocols have been widely used in adipose tissue clearing. However, benzyl alcohol/benzyl benzoate (BABB)-based clearing has been shown to offer better transparency in other tissues. The addition of diphenyl ether (DPE) to BABB (BABB-D4) is often included to preserve fluorescent signals, but its effects on adipose tissue transparency and shrinkage have not been explored. Distinct adipocyte subpopulations contribute to its cellular composition and biological activity. Here, we compared clearing solvents to create an optimized clearing methodology for the study of adipocyte subpopulations. Adipose tissues were cleared with BABB, BABB-D4, and DBE, and post-clearing transparency and tissue shrinkage were measured. An optimized protocol, including BABB-D4 clearing, delipidation, and extensive immunofluorescence blocking steps, was created to examine the spatial distribution of Wt-1 positive progenitor-derived (Type-1) adipocytes in intact mesenteric fat. Both BABB and BABB-D4 lead to significantly increased tissue transparency with reduced tissue shrinkage compared to DBE-cleared adipose tissue. Type-1 adipocytes are found in a clustered distribution with predominant residence in fat associated with the ileum and colon. This paper details an optimized clearing methodology for adipose tissue with increased tissue transparency and reduced shrinkage, and therefore will be a useful tool for investigating adipose tissue biology.

## 1. Introduction

Traditional immunohistochemistry approaches rely on mechanical sectioning and analyses of small sections, which limit the interpretation of 3D cellular microenvironments. Advances in imaging technology such as confocal and multiphoton microscopy replace mechanical sectioning with optical sectioning. Focus stacking of images taken by confocal microscopy has been widely used to study adipose structures in physiological and pathological conditions; however, the depth of visualization is usually only a few hundred micrometers due to refractive index mismatch [1]. Thus, optical clearing methods have been developed to match refractive indices of tissues to the surrounding media and enable deep-tissue imaging.

Three clearing method types (aqueous-, hydrogel-, or solvent-based) each come with advantages and disadvantages. Aqueous-based clearing methods provide biocompatibility, biosafety, and high degree of fluorescence preservation [2,3]. Nonetheless, they have reduced optical clearing power compared to other techniques [4]. Hydrogel-based methods offer benefits including increased tissue transparency and sample porosity, which allow penetration of antibodies; however, these require expensive equipment and complex tissue processing [2,5]. Solvent-based methods are fast and provide high degree of tissue transparency, but quickly quench fluorescence and require utilization of toxic chemicals [1].

White adipose tissue is broadly classified into subcutaneous and visceral, with the latter being associated with increased risk of metabolic diseases. Interestingly, not all visceral or subcutaneous adipocytes behave the same as evidenced by mesenteric adipocytes having replicability and differentiation capacity between abdominal subcutaneous and omental fat [6], suggesting the existence of intrinsic, cell-autonomous differences that contribute to the regional variations in mature adipocytes. Consistent with the idea of depot-specific metabolic differences, a growing body of evidence demonstrates that even within a fat depot, adipocytes are derived from multiple cellular lineages. Using clonal analysis and lineage tracing, our lab previously identified at least three subpopulations distinguished by the expression of marker genes Wilms’ tumor-1, Transgelin, Myxovirus-1, termed Type-1, Type-2, Type-3 adipocytes, respectively. These adipocyte subpopulations exhibit differential gene expression, metabolic properties, differential responses to exogenous stimuli including inflammatory cytokines, and differ in their distribution within individual adipose depots of a mouse [7].

To investigate spatial distribution of adipocyte subpopulations, previous studies have relied on marking the orientation of the collected fat pad and noting internal structures within the fat, such as the lymph nodes in inguinal subcutaneous fat [8,9,10]. However, the traditional dissection disrupts the relationship between mesenteric fat and the underlying intestines and mesentery; thus, optical clearing is required to observe cellular distribution in this depot. To date, solvent-based clearing has been used primarily to investigate vascularity [11] and sympathetic innervation in adipose browning processes [12,13,14]. The common clearing reagent used in these studies is dibenzyl ether (DBE), which in other tissues has been shown to have lower degree of tissue transparency and higher percentage of tissue shrinkage compared to BABB-based clearing [4,15]. Therefore, in this study, we compared DBE, BABB, and BABB-D4 (with added diphenyl ether for fluorescence preservation) clearing solvents on adipose tissue transparency and shrinkage, and found that BABB-based clearing agents significantly increase adipose tissue transparency and reduce tissue shrinkage. Thus, utilizing a BABB-based clearing reagent, delipidation, and extensive immunofluorescence blocking has allowed us to determine an optimized clearing protocol to investigate spatial distribution of Type-1 adipocytes in mesenteric fat. We found that Type-1 adipocytes were highly concentrated in fat regions associated with the ileum and colon, but to a lesser extent in the duodenum and jejunum, and that Type-1 adipocytes exhibited a clustered distribution within these regions of mesenteric fat. Thus, this methodology provides a direct improvement upon previously published clearing protocols in adipose tissue, and further describes a novel spatial/structural distribution of an adipocyte subpopulation in mesenteric fat.

## 2. Materials and Methods

Animals: Transgenic mice to label Type-1 adipocytes were generated by crossing Wt1-Cre^ERT2^ and Rosa26^mTmG^ [16] mice on a C57BL/6 background (Jackson Labs). Animals were housed at 22 °C, under a 14-h light/10-h dark cycle, with 4–5 mice per cage with ad libitum access to water and standard laboratory chow: 22% calories from fat, 23% from protein, and 55% from carbohydrates (ProLab RMH 3000, Lab Diet, St. Louis, MO, USA). Tamoxifen (40 mg/mL) was used to induce Cre at E 14.5 days using an 18G feeding needle with 2.4 mm smooth ball on the tip. Animal care and study protocols were approved by the Ohio University Institutional Animal Care and Use Committee and were in accordance with the National Institute of Health guidelines.

Collection: Six-week-old Wt1-Cre^ERT2^/Rosa26^mTmG^ mice were fasted overnight and were euthanized with CO_2_. Intestines and mesenteric fat tissues were carefully dissected at the level of pancreas and rectum. Intestines were gently flushed with 1X PBS using 23G needle to remove fecal content. A small amount of fecal matter was kept in the colon to further differentiate this region easily after clearing. However, fecal matter should be kept to minimum as it could dislodge and cause unnecessary autofluorescence in other parts of the tissues during imaging. The tissues were fixed with 4% paraformaldehyde at 4 °C overnight and were washed with 1X PBS three times 1-h each.

Delipidation and permeation: Samples were delipidated and permeabilized with dichloromethane (DCM-270997, Sigma Aldrich, St. Louis, MO, USA) and methanol, respectively, as suggested by the Adipo-Clear protocol [14,17]. Samples were washed in 20%, 40%, 60%, 80% methanol/B1N buffer (H2O/0.1%Triton X-100/0.3M glycine), and finally in 100% methanol for 30-min each. Tissues were placed in 100% DCM for delipidation for 30 min 3X. Samples were then washed with 100% methanol 30 min twice, then with 80%, 60%, 40%, and 20% methanol/B1N for 30 min each. All incubations were performed in 4 °C. Samples were washed with B1N for 30 min twice and with PTwH buffer (PBS/0.1% Triton X-100/0.05% Tween 20/2 µg mL^−1^ heparin) for 1 h twice.

Immunohistochemistry: Samples were blocked (PBS/0.2% TritonX-100/10%DMSO/10% Goat serum) and stained with primary (1:200 Chicken anti-GFP; GFP-1010, AVES Labs, Inc., Davis, CA, USA) and secondary (1:400 anti-Chicken Alexa Fluor 647; ab150171, Abcam, Cambridge, MA, USA) antibodies in PTwH for 4 days at 37 °C. Alexa Fluor 647 was used to avoid autofluorescence. Samples were washed with PTwH at RT during these intervals (15 min, 30 min, 1 h, 2 h, 4 h, and overnight) after primary and secondary antibody incubation. During incubation of these steps, tissues were rotated to facilitate diffusion of antibodies. Mesenteric fat and the intestines must be in a 5 mL falcon tube to avoid excessive mechanical tearing which can disrupt the integrity of the tissues before clearing.

Dehydration and clearing: Samples were dehydrated in 25%, 50%, 75%, 100% methanol/H_2_O for 30 min each at RT. Samples were then washed with 100% DCM for 30 min. They were placed in BABB-D4 solution overnight in the dark before proceeding to imaging. BABB-D4 consisted of 1:2 benzyl alcohol (305197, Sigma Aldrich, St. Louis, MO, USA): benzyl benzoate (B6630, Sigma Aldrich, St. Louis, MO, USA) (BABB); 1:4 diphenyl ether (A15791, Alfa Aesar, Haverhill, MA, USA): BABB [18].

Imaging: Samples were then imaged using confocal microscopy (NIKON A1R, Nikon Instruments Inc., Melville, NY, USA) while submerged in BABB:D4 solution. The imaging chamber was a 60 mm glass dish with a 25 mm bottom opening. Analyses were conducted using confocal NIKON A1R analysis software. Images of the duodenum, jejunum, ileum, and colon from both male and female mice were taken using the same setting and setup (10X objective and 1.0 µm pinhole).

Solvent comparisons: Approximately 1 × 1 cm^2^ perigonadal and inguinal subcutaneous tissues of two mice were cleared using different organic solvents (BABB, BABB-D4, and DBE), *n* = 8/group. To compare the level of post-clearing transparency, tissue absorbance across the visible light spectrum (350–700 nm) was read using a 96-well plate reader (BIOTEK Gen5, Winooski, VT, USA). Each sample was read three times, and an average absorbance value was obtained for analysis. Tissue area and thickness were measured before and after clearing to determine volume change. Perigonadal and subcutaneous fat were tested for their transparency and volume change 2 days after clearing.

Analysis: Concentration of GFP+ adipocytes for each region of mesenteric fat was determined by calculating the percentage of volume occupied by GFP+ adipocytes over the volume of fat, i.e., excluding intestines/pancreas and empty space captured in a particular frame. All data were analyzed using ANOVA followed by Tukey’s post hoc analysis. GFP+ adipocyte distribution was analyzed using Kernel density estimation (KDE), which estimates the spatial distribution probability of populations of interest. The statistical estimate uses stat_density2d function which assumes Kernel function to be Gaussian with diagonal bandwidth matrix [19,20]. The KDE script was written, run, and analyzed in R Studio (see Appendix A).

## 3. Results

### 3.1. BABB and BABB-D4 Clearing Increased Adipose Tissue Transparency and Reduced Tissue Shrinkage Compared to DBE-Based Clearing

Previous studies have shown that BABB is generally more robust at clearing in other tissues than DBE due to higher transparency and lower size reduction [4,15]. However, because most adipose tissue protocols utilize DBE as the clearing agent, direct comparison of these two reagents in adipose tissue is needed. The addition of diphenyl ether (DPE) to BABB at ratio 1:4 (BABB-D4) has been shown to preserve GFP fluorescence in other tissues [18]. Therefore, in this study, we compared the relative effects on transparency and size reduction in adipose tissue of DBE, BABB, and BABBB-D4 solvents. As perigonadal and inguinal subcutaneous fat possess different properties, we investigated the effects of these organic solvents on both groups.

Visual inspection of post-cleared samples suggested that tissues in BABB and BABB-D4 were more transparent compared to those in DBE for both perigonadal and subcutaneous fat (Figure 1A). To confirm this finding, we quantified the degree of transparency by measuring the absorbance of individual samples in their respective organic solvent in a 96-well plate. Tissues were read at various wavelengths in the visible spectrum (350–700 nm), with higher absorbance values indicating less transmittance of light to the detector and a lower degree of transparency. Both BABB and BABB-D4 led to significantly increased tissue transparency compared to DBE at wavelengths between 370 and 490 nm for perigonadal fat (Figure 1B,D) and at all measured wavelengths for subcutaneous fat (Figure 1C).

To determine the degree of tissue size reduction caused by each solvent, the volume of each sample, before and after clearing, was calculated from the measured tissue thickness and area, with positive and negative percentages representing reduction and expansion, respectively (Figure 2A). BABB (−21 ± 19%, subcutaneous, 13 ± 6% visceral) and BABB-DE (20 ± 4% subcutaneous; 20 ± 5% visceral) led to lower tissue volume reduction compared to that of DBE (37 ± 4% subcutaneous; 24 ± 6% visceral). However, this only reached statistical significance in subcutaneous fat (Figure 2B,C). Thus, BABB-based clearing leads to reduced tissue shrinkage compared to DBE-based clearing.

### 3.2. Type-1 Adipocytes Are Spatially Clustered and Are Preferentially Concentrated in Mesenteric Fat Associated with the Ileum and Colon

Previously, our lab and others have performed lineage tracing analysis utilizing Wt1-Cre^ERT2^ mice crossed with mice carrying the membrane Tomato fluorescent protein (mTFP)/membrane green fluorescent protein (mGFP) reporters expressed in the ROSA26 locus to label a subset of visceral adipocytes, which we termed Type-1 adipocytes [7]. In cells expressing the Cre recombinase and all future cell lineages derived from these cells, there is gene rearrangement resulting in a loss of mTFP and an expression of mGFP. In this study, our goal was to utilize a solvent-based clearing technique to investigate the spatial distribution of mGFP+ Type-1 adipocytes in the mesenteric fat of Wt1-Cre^ERT2^/ROSA26^mTmG^ mice.

Mesenteric fat and associated intestines were successfully cleared using BABB-D4 solution (Appendix A). Organic solvent clearing quenches endogenous signals, thus requiring the utilization of exogenous fluorescent antibodies to increase fluorescent protein detection. However, antibody penetration was limited as seen by the localization of GFP+ fluorescence on the outer edge of tissue (Figure 3A). Nonetheless, using the same immunohistochemistry steps, successful staining of sympathetic nerves and blood vessels was observed with anti-tyrosine hydroxylase and isolectin-B4, respectively (Appendix A). A potential problem could arise from the abundance of hydrophobic lipid, which hindered the diffusion of antibodies and impaired proper staining. Dichloromethane (DCM) was used to remove lipid from adipose tissues prior to immunolabelling, as suggested by the Adipo-Clear protocol [14,17]. DCM improved penetration of anti-GFP antibodies into tissues and increased GFP signal, but also led to high background fluorescence (Figure 3B). A significant reduction in background fluorescence was observed by the addition of extensive blocking of tissues in goat serum, which allowed better visualization of individual Type-1 adipocytes (Figure 3C). Autofluorescence emitted from the tissue itself was not blocked by goat serum and was utilized to locate the positions of GFP+ adipocytes relative to other unstained structures (e.g., blood vessels and intestines) (Figure 3D). Similar results were obtained in the perigonadal fat (Figure 3E). The GFP-labelled cell membrane is more obvious in adipocytes of perigonadal fat because of the larger size. Additionally, depth analysis, represented by various colors, showed that Type-1 adipocytes were observed throughout mesenteric fat (Figure 3F), suggesting antibodies penetrated throughout tissue thickness rather than only the tissue surface as seen in Figure 3A.

Type-1 adipocytes displayed a proximal–distal distribution, with proximal and distal referring to areas closest to the duodenum and rectum, respectively (Figure 4A). Intact solvent-cleared mesenteric fat allowed quantitative analyses of GFP+ adipocytes at different sites along the intestinal tract. Normally, the percentage of GFP+ adipocytes is used as a quantitative comparison among fat depots or regions; however, this method is not reliable as the total number of fat cells could not be fully assessed in these studies. Rather, percentage of occupied space by GFP+ adipocytes (volume) to the total fat volume in a particular image frame was calculated. The calculation considers both the size and number of GFP+ adipocytes. Because Type-1 adipocyte sizes (in volume) did not vary across regions (Appendix A), a higher percentage of occupied volume of these adipocytes in a region indicated a higher concentration of these cells relative to other regions. On average, the volume occupied by GFP+ adipocytes in the ileum and colon were ~3 to 5-fold higher than those in duodenum and jejunum (0.4 ± 0.09% and 0.3 ± 0.04% compared to 0.07 ± 0.03% and 0.09 ± 0.03%, respectively) (Figure 4B). These results were confirmed by conventional confocal microscopy of mesenteric fat, which showed, prior to clearing, that Type-1 adipocytes in the ileum and colon were 11 and 12-fold higher than in the duodenum, respectively (Appendix A).

Kernel density estimation (KDE), a statistical method used to estimate spatial distributions, also revealed that Type-1 adipocytes were spatially clustered in the ileum and colon (Figure 4C). Density values with arbitrary unit vary depending on the input data (i.e., the number of GFP+ adipocytes and their respective locations in a 2D plane). Cellular density is represented by a color gradient, with red being the most concentrated clusters. Higher density indicates higher probability of finding GFP+ adipocytes in that region.

## 4. Discussion

The current protocols on volumetric imaging of fat focus on sympathetic innervation and vasculature [11,12,13,21]. All previously published adipose tissue clearing studies employed DBE as the clearing reagent [17,22,23]. However, comparing to DBE, BABB provided increased adipose tissue transparency [4]. The degree of transparency is consistently better in both subcutaneous and visceral fat between 370 and 490 nm, suggesting that this protocol would increase sensitivity in all adipose tissue protocols that utilize commonly used red and green fluorophores. Additionally, tissue shrinkage was significant in DBE-cleared tissues, leading to possible deformation and spatial distortion [15]. Our results showed a trend of lower size reduction in BABB compared to DBE, suggesting that BABB is potentially a better reagent for studies that rely on tissue integrity (e.g., spatial distribution of adipocyte subpopulations). However, studies seeking to shrink large samples for ease of imaging may consider DBE.

In this manuscript, we propose an optimized protocol that utilizes BABB-D4 as the clearing reagent, utilizes DCM-based delipidation as described in the Adipo-Clear protocol [17], and adds extensive immunofluorescence blocking of tissue in goat serum to achieve increased tissue transparency, reduced tissue shrinkage, and greater signal-to-noise ratio compared to those achievable with previous published protocols. Although BABB was originally used with ethanol, hexane [24], or tert-butanol [18], this study also shows that BABB is compatible with methanol and DCM for maximal dehydration, permeation, and delipidation of adipose tissues. Taken together, these results indicate that BABB-D4-based protocol represents a significant improvement upon previously published protocols and will be a useful tool for investigating adipose tissue biology.

The more commonly used microscopy with solvent-clearing is light-sheet fluorescence microscopy which has its advantages including rapid imaging speed and low photobleaching and phototoxic effects [25]. Despite the advantages, light-sheet microscopy is not readily available for all users compared to confocal microscopy. While confocal microscopy can only visualize tissue surface ~100 µm in adipose tissue in a pre-clearing state, it can reach to the depth of a millimeter in cleared samples. Although the need to submerge samples into organic solvents during imaging can damage confocal objective lenses, the construction of a simple glass chamber with thin glass slides to cover the aperture and to reduce the objective distance in inverted confocal microscope will suffice (Figure 5).

Adipocytes derived from visceral mesothelium, Type-1 adipocytes, were detected in visceral but not subcutaneous depots [7,26]. Type-1 preadipocytes and adipocytes contribute approximately 5% and 6% to the mesenteric depot, respectively [7]. In addition to making up only a small percentage of mesenteric depot, these adipocytes were non-randomly distributed and were predominantly found along the distal intestinal tract (ileum and colon). The exact physiology of such distribution is unknown and warrants further investigation. We have previously shown that adipocyte subpopulations differentially respond to inflammatory cytokines [7]. Because inflammatory bowel diseases are commonly seen in the terminal ileum and colon where Type-1 adipocytes predominantly reside, we speculate that distribution of these adipocytes may, at least in part, influence immune properties of the whole tissue.

Tissue heterogeneity is not limited to adipose tissues. The development of single-cell RNA sequencing coupled with computational analysis has begun to uncover molecular and functional heterogeneity. In fact, both BABB and DBE reagents have been shown to clear tumor tissues [22,27]. Heterogeneity of cancer cells and spatial distribution of T cells may dictate tumor response to anti-cancer therapies [28]. Therefore, optical clearing can be an alternative tool for visualizing and analyzing spatial distribution of various cellular subpopulations in tissues.

We have demonstrated the unique spatial distribution of mesothelium-derived Type-1 adipocytes in intact mesenteric fat using BABB-D4 clearing method. This methodology directly improves previously published protocols, can be utilized to study spatial distribution of other cell types within adipose tissue, and will be a useful tool for investigating adipose tissue biology in general. Furthermore, this protocol can be readily adapted to examine the function and spatial relationship of heterogeneous cell populations in other tissues and cancers.

## Figures and Tables

**Figure 1 mps-04-00039-f001:**
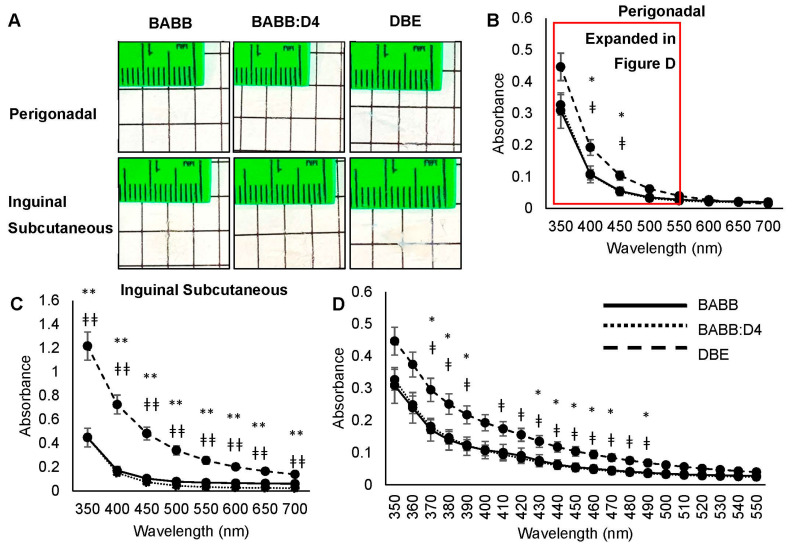
Perigonadal and inguinal subcutaneous adipose tissues are more transparent in BABB +/− DPE solvents compared to DBE. (**A**) Representative images of approximately 1 cm × 1 cm fat tissue in their respective solvents. Comparing absorbance values from 350 to 700 nm among BABB +/− DPE and DBE clearing solutions of (**B**) perigonadal and (**C**) inguinal subcutaneous samples. (**D**) Comparing absorbance values from 350 to 550 nm among BABB +/− DPE and DBE clearing solutions. *n* = 8, * <0.05 BABB vs. DBE, ‡ <0.05 BABB-D4 vs. DBE, ** <0.001 BABB vs. DBE, ‡‡ <0.001 BABB-D4 vs. DBE.

**Figure 2 mps-04-00039-f002:**
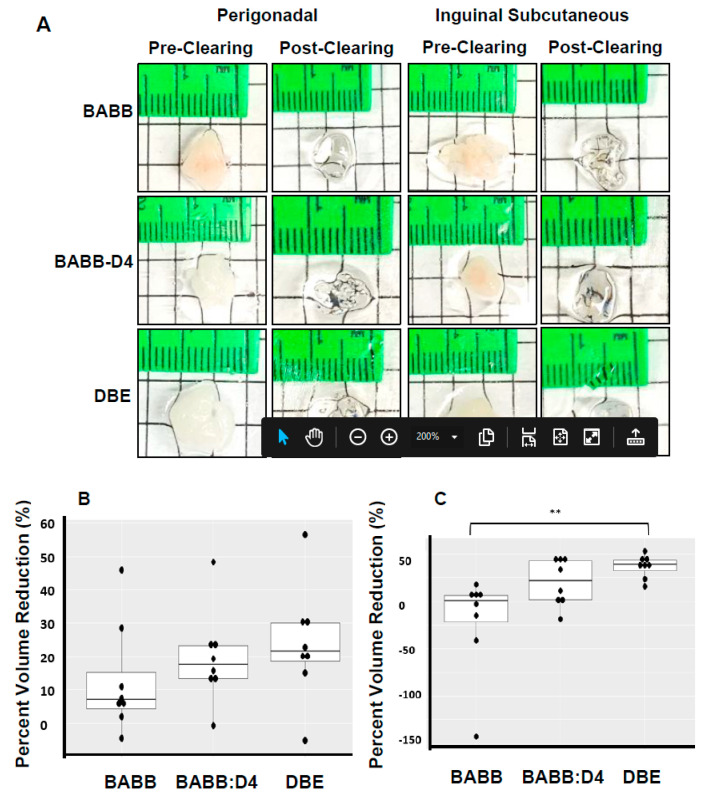
Three organic solvents have differential effects on tissue size. (**A**) Representative images between pre-and post-clearing of perigonadal vs. inguinal subcutaneous in three different organic solvents. Percent size change in (**B**) perigonadal and (**C**) inguinal subcutaneous fat. *n* = 8, ** *p* < 0.01.

**Figure 3 mps-04-00039-f003:**
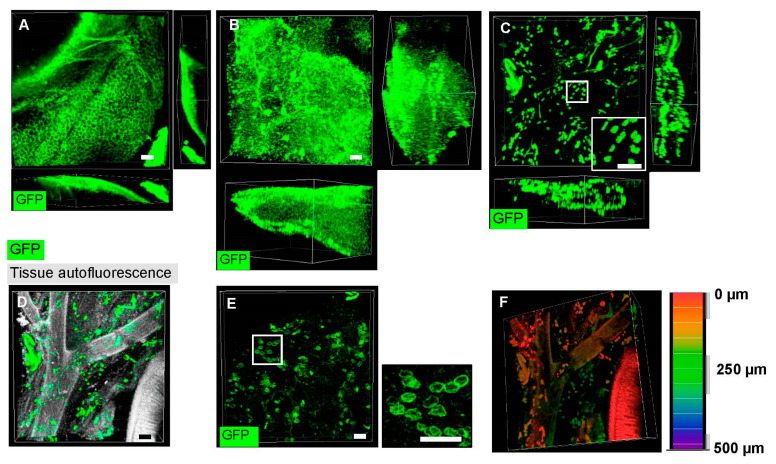
Development of optimized clearing protocol including BABB-D4 based clearing, DCM delipidation, and increased blocking improves staining of Type-1 adipocytes in mesenteric adipose tissue associated intestines of Wt1-cre^ERT2^/ROSA2^6mTmG^ mice. Representative confocal images (and relevant orthogonal view of tissues) of BABB-D4-cleared tissue (**A**) without DCM delipidation and (**B**) with DCM delipidation; (**C**) Representative confocal image of tissue cleared with the final clearing protocol including DCM delipidation, BABB-D4-clearing tissue, and increased blocking. An enlarged image of GFP+ adipocytes are shown in the right bottom corner; (**D**) Image C with tissue autofluorescence (gray) to observe the positions of GFP+ adipocytes with respect to other tissue structures (e.g., intestines and blood vessels); (**E**) Representative confocal image of BABB-D4-cleared perigonadal tissues with more obvious and larger adipocytes; (**F**) Color-coded depth representation of image C, demonstrating the penetration of primary anti-GFP and secondary Alexa 647 conjugated antibodies throughout tissues. All pictures were taken at 100X magnification (10X objective). Scale bar = 100 µm, *n* =2 mice ~6 weeks of age.

**Figure 4 mps-04-00039-f004:**
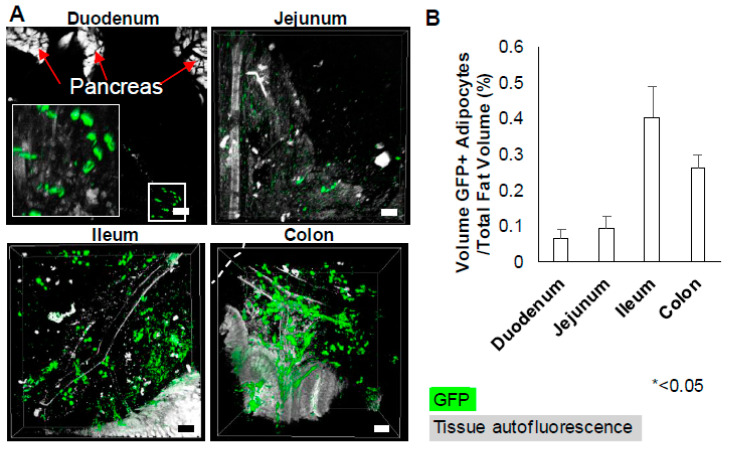
Type-1 adipocytes are spatially distributed in adipose tissues around the ileum and colon in Wt1-cre^ERT2^/ROSA26^mTmG^ mice. (**A**) Type-1 adipocytes are observed in greater number in adipose tissues associated with ileum and colon. (**B**) Quantitative analysis of percentage of occupied volume of Type-1 GFP+ adipocytes to the overall fat volume per frame. Data are shown as mean ± SEM of two ~6-week-old mice, four frames/mouse. (**C**) Representative images of mesenteric fat around the ileum and colon, and Kernel density estimation (KDE) demonstrate the non-random clustering distribution of Type-1 adipocytes in these regions. KDE estimate the probability of finding Type-1 cells is higher in red areas and lower in yellow areas. Blue vessels are a digitized depiction of vessels based on the autofluorescence from the tissues. All pictures were taken at 100x magnification. Scale bar = 100 µm. In all images, green indicates Alexa 647 to detect anti-GFP signals, while the gray represents tissue autofluorescence. *p* < 0.05; two-way ANOVA; * <0.05.

**Figure 5 mps-04-00039-f005:**
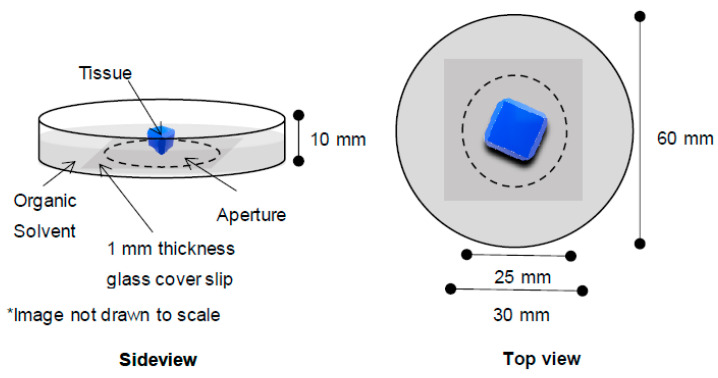
Imaging chamber for cleared tissues using inverted confocal microscopy. Epoxy glue was used to adhere the glass cover slip to the bottom of the glass dish. The epoxy glue resisted erosion from organic solvents in the tested duration (72 h). Not drawn to scale.

## Data Availability

Not applicable.

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
