# Peer review of "Development of an Optimized Clearing Protocol to Examine Adipocyte Subpopulations in White Adipose Tissue"

_mps, 2021, doi:10.3390/mps4020039_

Round 1

Reviewer 1 Report

In the present paper, the authors compared clearing solvents to create an optimized clearing methodology for the study of adipocyte subpopulations. Adipose tissues were cleared with BABB, BABB-D4, and DBE, and post-clearing transparency and tissue shrinkage were measured.

I consider this paper report original and well written. The authors tested an optimized clearing methodology for adipose tissue with increased tissue transparency and reduced shrinkage and the protocol developed is feasible, reproducible, and very useful in investigating adipose tissue biology. 

Author Response

We would like to thank the reviewer for their time and attention to our manuscript. We are delighted by the positive review and that the reviewer shares our belief that our optimized clearing protocol for adipose tissue will be a significant contribution to the field.

Reviewer 2 Report

In the manuscript "Development of an Optimized Clearing Protocol to Examine Adipocyte Subpopulations in White Adipose Tissue", Luong et al. described a clearing methodology for 3-d imaging of adipose tissue. It is a useful contribution to the field. However, the manuscript would be improved if multi-channel co-staining could be showed. For example, in Fig. 3 the tomato channel of mTmG mice derived tissue should be presented for better evaluation of type-1 adipocytes.

Author Response

Point:  It is a useful contribution to the field. However, the manuscript would be improved if multi-channel co-staining could be showed. For example, in Fig. 3 the tomato channel of mTmG mice derived tissue should be presented for better evaluation of type-1 adipocytes.

Response: We would like to thank the reviewer for their time and attention to our manuscript. We are delighted that they share our belief that this protocol will be a useful contribution to the field. Although we agree with the reviewer that the addition of tomato fluorescence would be helpful in the evaluation of GFP+ Type 1 adipocytes, these fluorescence signals are not available. Organic solvent clearing methods quench endogenous fluorescent signals, and immunofluorescence labeling is needed to detect protein/cell of interest. However, we are confident in these results of the study, as we do show that the remainder of the non-GFP adipocytes are Tomato+ (Supplemental Figure 3).